# Obesity Risk-Factor Variation Based on Island Clusters: A Secondary Analysis of Indonesian Basic Health Research 2018

**DOI:** 10.3390/nu14050971

**Published:** 2022-02-24

**Authors:** Sri Astuti Thamrin, Dian Sidik Arsyad, Hedi Kuswanto, Armin Lawi, Andi Imam Arundhana

**Affiliations:** 1Department of Statistics, Faculty of Mathematics and Natural Science, Universitas Hasanuddin, Makassar 90245, Indonesia; hedikuswanto454@gmail.com; 2Department of Epidemiology, Faculty of Public Health, Universitas Hasanuddin, Makassar 90245, Indonesia; sidik@unhas.ac.id; 3Department of Cardiology, Division of Heart and Lungs, University Medical Centre Utrecht, University of Utrecht, 3584CS Utrecht, The Netherlands; 4Department of Mathematics, Faculty of Mathematics and Natural Science, Universitas Hasanuddin, Makassar 90245, Indonesia; armin@unhas.ac.id; 5Department of Nutrition, Faculty of Public Health, Universitas Hasanuddin, Makassar 90245, Indonesia; andi.imam@unhas.ac.id; 6Central Clinical School, Faculty of Medicine and Health Science, The University of Sydney, Sydney 2050, Australia

**Keywords:** body weight, Indonesia, islands cluster, multiple logistic regression, obesity, risk factor

## Abstract

Obesity has become a rising global health problem affecting quality of life for adults. The objective of this study is to describe the prevalence of obesity in Indonesian adults based on the cluster of islands. The study also aims to identify the risk factors of obesity in each island cluster. This study analyzes the secondary data of Indonesian Basic Health Research 2018. Data for this analysis comprised 618,910 adults (≥18 years) randomly selected, proportionate to the population size throughout Indonesia. We included 20 variables for the socio-demographic and obesity-related risk factors for analysis. The obesity status was defined using Body Mass Index (BMI) ≥ 25 kg/m^2^. Our current study defines 7 major island clusters as the unit analysis consisting of 34 provinces in Indonesia. Descriptive analysis was conducted to determine the characteristics of the population and to calculate the prevalence of obesity within the provinces in each of the island clusters. Multivariate logistic regression analyses to calculate the odds ratios (ORs) was performed using SPSS version 27. The study results show that all the island clusters have at least one province with an obesity prevalence above the national prevalence (35.4%). Six out of twenty variables, comprising four dietary factors (the consumption of sweet food, high-salt food, meat, and carbonated drinks) and one psychological factor (mental health disorders), varied across the island clusters. In conclusion, there was a variation of obesity prevalence of the provinces within and between island clusters. The variation of risk factors found in each island cluster suggests that a government rethink of the current intervention strategies to address obesity is recommended.

## 1. Introduction

Obesity is a major public health issue causing multiple burdens of co-morbidities and mortalities among adults. The World Health Organization (WHO) reported that globally 39% of adults were overweight and 13% were obese, and this number has nearly tripled within the last three decades [1]. In Indonesia, the obesity prevalence has increased significantly from 18.8% in 2007 to 26.6% in 2013, with a slight decrease in 2018 (21.8%) [2,3,4].

There are many statistical methods for analyzing large-scale study data. The machine learning method is a powerful statistical analysis approach that can be used for predictive model development of health outcomes. A recent systematic review reported various machine learning techniques that were performed to predict adult obesity from nationwide and large cross-sectional data, finding that logistic regression analysis had the highest accuracy in predicting obesity [5,6]. This finding is in line with our previous study [7], which found that logistic regression had the highest performance in predicting and measuring obesity. Predicting obesity risk factors by considering determinant variables can be advantageous to design and modify local existing nutrition programs and policies better for controlling the obesity problem.

To the best of our knowledge, this is the first study re-analyzing cross-sectional Indonesian Basic Health Research (RISKESDAS in Indonesian) data based on the main islands in Indonesia (we use the term “island clusters”). A previous study in Indonesia investigating the determinants of obesity among adults using the 2007 and 2013 RISKESDAS data concluded that the prevalence of obesity and risk factors varied among the areas [8]. However, this study only grouped the areas based on Indonesia’s three different time regions, which might cause bias within the three groups. Therefore, further analysis for obesity determinants in regions with similar population characteristics is essential to minimize the variation bias. We clustered the provinces located on the same island into one cluster as the population characteristics within the same island cluster, assuming that within-island populations share more characteristics than clusters determined only by the time zone.

The main aim of this study is to examine the factors contributing to obesity in adults and investigate how these varied across the island clusters. This study also describes the prevalence of obesity in seven island clusters in Indonesia and reveals what factors increase or decrease the risk of obesity.

## 2. Materials and Methods

### 2.1. Data Source

Secondary data analysis performed in the current study was based on the data from the RISKESDAS study, a nationally representative cross-sectional study in Indonesia conducted by the Ministry of Health in 2018. Detailed information regarding methods, ethical considerations, and other related aspects of the RISKESDAS study is published elsewhere [9]. Briefly, the RISKESDAS sample was selected based on 2010 population census blocks using multi-stage cluster random sampling. Our data for analysis comprised 618,910 adults (≥18 years) from approximately 300,000 households randomly selected using proportionate to population sub-samples throughout Indonesia [2]. 

The data can be obtained from the National Institute of Health Research and Development (NIHRD), Ministry of Health, Republic of Indonesia upon request (https://www.litbang.kemkes.go.id/layanan-permintaan-data-riset/, accessed on 3 May 2021).

### 2.2. Study Variables

Socio-demographic variables, obesity status, and selected risk factors were identified from RISKESDAS 2018 questionnaires prior to the data request. We included 20 variables for socio-demographic and obesity-related risk factors for analysis. The socio-demographic variables of sex, education, employment, marital status, and urban-rural status were included. 

Obesity status was calculated based on the Body Mass Index (BMI) using weight and height. We classify an individual as obese with a BMI ≥ 25 kg/m^2^ following WHO BMI cut-offs for Asian populations [10]. Mental and Emotional Disorders (MEDs) were based on 20 items from a Self-Reporting Questionnaire (SRQ) developed by the WHO [11]; we determined a MED with the cut-off point ≥6 (positive predictive value = 70%, and negative predictive value = 92%) [12]. The frequency of consumption of risky food items (sweet foods and beverages, high-in-salt foods, high-in-fat foods, meats, carbonated drinks, energy drinks, and instant foods) was measured. The eating or drinking of risky food items more than 1 time per day was considered as high frequency consumption. Vegetable and fruit consumption was calculated based on the WHO standard [13]; ≥5 portion per day was determined as adequate. Smoking status was classified as “currently smoking”, “quit smoking”, and “never smoked”, based on participant self-reporting. Physical activity in the current analysis was based on the WHO Global Physical Activity Questionnaire (GPAQ) used in the RISKESDAS study; sufficient physical activity was defined according to the WHO recommendations [14]. Drinking alcoholic beverages within 1 month prior to the study was defined as alcohol consumption. Blood pressure was measured using systolic and diastolic blood pressure during data collection; the 8th Joint National Committee guideline was used to classify blood pressures [15]. 

### 2.3. Island Clusters 

Indonesia is the largest archipelagic country consisting of clusters of islands divided into 34 provinces. Our current study defined 7 major island clusters: Sumatra (provinces included: Aceh, North Sumatra, West Sumatra, Riau, Jambi, South Sumatra, Bengkulu, Lampung, Kepulauan Bangka Belitung, and Kepulauan Riau); Java (provinces included: DKI Jakarta, West Java, Central Java, Yogyakarta, East Java, and Banten); Bali-Nusa Tenggara (provinces included: Bali, West Nusa Tenggara, and East Nusa Tenggara); Kalimantan (provinces included: West Kalimantan, Central Kalimantan, South Kalimantan, East Kalimantan, and North Kalimantan); Sulawesi (provinces included: South Sulawesi, Central Sulawesi, Southeast Sulawesi, North Sulawesi, West Sulawesi, and Gorontalo); Maluku (provinces included: Maluku and North Maluku), and Papua (provinces included: Papua and West Papua). 

### 2.4. Statistical Analysis

The sample weights for the complex survey design were considered in the analysis. Descriptive analysis was conducted to determine the characteristics of the population and to calculate the prevalence of obesity within the provinces in each island’s cluster. 

In order to calculate the adjusted odds ratios (ORs), multivariate logistic regression analyses, which includes other variables associated with obesity, were performed. The selection of multivariate logistic regression to develop a predictive model was based on our prior study that showed a high-performance, including accuracy, specificity, precision, Kappa, and $F_{\beta }$ [7]. Multivariate logistic regression was performed using SPSS version 27 (IBM Corp, Armonk, NY, USA). Cohen’s and Cliff’s Delta analyses were performed using R version 4.0.1 (‘effsize’ package version 0.7.6, Marco Torchiano, 2019) to validate each factor’s variation (effect size) of odds ratios (ORs) by island cluster. The effect sizes were presented in four distinct categories: negligible, small, medium, and large. The effect size was considered negligible if the score was below 0.2, a score of 0.2 to <0.5 was considered small, 0.5 to <0.8 for medium, and 0.8 and above for a large effect size [16,17].

### 2.5. Ethical Considerations

Ethical approval of the RISKESDAS survey was obtained from the Ethical Committee of Health Research, the Indonesian Ministry of Health (LB.02.01/2/KE.267/2017) [2].

## 3. Results

### 3.1. Prevalence of Obesity across Island Clusters

The purpose of this study was to describe the prevalence of obese adults by island cluster in Indonesia and assess the risk of obesity caused by determinant factors using the secondary data of RISKESDAS 2018. Figure 1 illustrates the estimated obesity prevalence distribution across Indonesia. The figure shows that all island clusters have at least 1 province in which the prevalence of obesity is more than 20%, and only 3 clusters with at least 1 province in which more than 25% are classified as obese: Java, Kalimantan, and Sulawesi Island.

In Sumatra, North Sumatra had the highest number of obese adults, while Lampung had the lowest (40.6% vs. 29.6%). The sequence of the regency for the highest and the lowest obesity prevalence in other island clusters were: DKI Jakarta (45.6%) versus Central Java (33.3%) in Java Island; Bali (38.8%) versus East Nusa Tenggara (19.1%) in Bali and Nusa Tenggara Island; East Kalimantan (44.1%) versus West Kalimantan (30.3%) in Kalimantan Island; North Sulawesi (46.6%) versus West Sulawesi (31.4%) in Sulawesi Island; Maluku Utara (37.9%) versus Maluku (33.0%) in Maluku Island; and West Papua (39.6%) versus Papua (35.0%) in Papua Island. DKI Jakarta had the highest proportion of an obese individuals, which outnumbered the national figure (45.6% vs. 35.4%), followed by Sulawesi Utara and East Kalimantan (data displayed in detail in the Appendix A).

Table 1 shows the breakdown of obesity prevalence in all the island clusters in Indonesia, according to categories of the determinants associated with obesity. Notably, this table indicates that obesity prevalence varied according to MED status. The distribution of obesity was higher among those without MED in the Sumatra, Java, and Papua Island clusters. In addition, a few variables show the variation of obesity prevalence by categories, including food high in salt, meat, carbonated beverage consumption, and smoking status. Individuals with a high level of education and having a permanent job (e.g., government/police/military officer) show a higher obesity prevalence than their counterparts.

### 3.2. Cluster Variation of Obesity Risk Factors

The results of the logistic regression analysis are displayed in Figure 2. The variables associated with increased odds (OR > 1) of being obese in all island clusters were the location (X01), gender (X02), marital status (X03), occupational status (X06), high-fat food (X11), and blood pressure (X20). Meanwhile, age group (X04), educational level (X05), sugar-sweetened beverage consumption (X09), FV consumption (X16), and smoking status (X17) factors statistically did not appear to increase the obesity risk in Indonesian adults.

These results suggest that six out of twenty variables show a variation among island clusters. For example, adults working as a government/military/police officer are over represented in the data, suggesting that employment in these sectors is one of the highest contributing factors to obesity in all clusters. However, working as an entrepreneur is the highest contributor to obesity levels in Bali and Nusa Tenggara Islands (OR = 1.775; 95% CI: 1.764–1.786). People with mental–emotional disorders are likely to be at risk of obesity only in Sumatra and Papua. Regarding smoking status, those who are still smoking tend to have a normal weight status compared to their counterparts. The OR number is displayed in more detail in Appendix A. In order to measure the effect size of each variable across the island clusters, we performed Cohen’s and Cliff’s Delta analyses (Appendix A).

## 4. Discussion

The present study found that some island clusters share common obesity risk factors, including individual and socio-economic factors. Individual factors (e.g., sex, high-fat food consumption, and blood pressure) and socio-economic factors (e.g., marital status, occupational status, and location) contributed to the risk of obesity in seven island clusters, indicating that these variables are probably strong predictors of obesity in Indonesia.

The present study showed that women have greater odds of being obese than men. Women typically have a body fat percentage around 10% higher than men [18,19]. Although other biological factors, such as age and ethnicity, also contribute to the adiposity distribution and percentage, women still have more considerable body fat in almost all life spans [18]. This result aligned with a previous study in a developing country that reported that the prevalence of obesity in adult females was higher than in males. The study also revealed that obesity was directly proportional to age, but only for females [20]. We also performed additional analyses to measure the risk of obesity among married women. The risk of obesity among women who were already married was likely to increase, but not for those living in the Ambon and Papua Island clusters. This may be due to socio-economic factors forcing women in these clusters to earn money [21], thus increasing metabolic energy expenditure.

In this study, high-fat food consumption (X11) was shown to be the only dietary factor that significantly contributed to the risk of obesity in Indonesian adults in all clusters. This finding was in line with a previous study, which found that the consumption of food containing a high-fat content was the risk factor of obesity in all regions in Indonesia and was consistently found in 2007 and 2013 [8]. These results are likely to be related to people’s eating habits differing by region. Eastern Indonesia tends to consume high-fat foods. It can be seen that the clusters of Sulawesi and Bali and Nusa Tenggara have a 1.4 times higher risk factor of being obese due to high-fat food consumption. Fat contributes significantly to the total energy intake, and thus reducing the high-fat food consumption might balance energy expenditure and intake [22]. Additionally, it should be noted that some fat types have beneficial effects on obesity. For example, replacing protein and short fatty acid (SFA) with polyunsaturated fatty acid (PUFA) has been shown to be significantly associated with a lower obesity risk [23].

Level of education (X5), age group (X4), sweet-sweetened beverages (SSBs) consumption (X9), energy drinks (X14), fruit and vegetable (FV) consumption (X16), and smoking status (X17), on the other hand, are statistically considered a lower obesity risk. Adults with low education levels have a lower risk of being obese than those with high education levels. This result is inconsistent with a larger cross-sectional study that found that the lower the years of education, the higher the odds of obesity [24]. What is surprising is that individuals that consumed sugar-sweetened and energy beverages had lower odds of being obese. This finding is contrary to many previous studies, which suggested that a greater intake of SSBs was associated with being overweight and obesity in children and adults [25,26]. This inconsistency might be due to the type of sugar contained in the beverages. Studies have shown that fructose-sweetened beverages increase adiposity levels more significantly than sucrose-sweetened beverages [25,27]. Unfortunately, we did not identify the dominant type of sugar contained in the beverages. Moreover, although low FV consumption reduces the risk of obesity, it shows a positive effect on the interaction between high-fat food and obesity. People with inadequate FV consumption are likely to consume more high-fat foods than those with adequate FV in-takes.

Adults who already quit smoking had increased odds of obesity in all the island clusters. This may be due to the effect nicotine has on the central nervous system and metabolism with two possible mechanisms. First, people who quit smoking tobacco tend to replace the hand-to-mouth smoking activity with eating, leading to an increase in calorie intake. Second, taste preference is also changed among those who quit smoking to obtain a pleasure “sensation” replacing the effect of tobacco [28]. However, this finding must be interpreted with caution because we have no baseline data to describe how much weight gain occurred after smoking cessation. Additionally, the use of tobacco can promote other diseases, such as cardiovascular diseases (CVDs), hypertension, and even mortality, the same risk posed to obese people [29]. Obesity and tobacco use do not actually show the opposite outcomes. Combining weight management and smoking cessation treatment might be promising in order to improve health quality and prevent the risk of metabolic diseases related to obesity and smoking behavior.

Interestingly, mental health disorders (X7), sweet-food consumption (X8), high-salt food (X10), meat consumption (X12), and carbonated drinks (X13) varied across the island clusters. Health-related behaviors might be different between the island clusters due to several factors, such as health inequalities, socio-economic status, or household deprivation [30,31]. The most obvious variation was carbonated drink consumption, which is a high-risk factor in Papua Island, but is not excessively consumed in Bali and Nusa Tenggara Island (OR = 1.81 vs. 0.84). It seems that this variation occurs because the adult diet in Bali and Nusa Tenggara is different from Papua. There is a great difference in OR values for other diet factors, such as high-salt food, high-fat food, and energy drink consumption. Similar to our findings, a study on children and adolescents found that the consumption of SSB (including carbonated drinks) varied by race/ethnicity, sex, and age [32]. 

Variation among clusters was also found for the mental health disorder factor (X7). This study shows that Papua Island had the highest risk of obesity caused by mental health disorders. Many conditions could trigger mental health issues, including the limitation of food choices, poor access to health services, high-risk behaviors, and poor education [33], all of which are challenges faced on Papua Island. However, there is no information about the source of mental health disturbance of the population. More extensive evidence investigating the roles of mental health in mediating obesity occurrence is needed.

We also noted that physical activity was a predictor of obesity in all island clusters. This result was consistent with many prior studies conducted in Indonesia and elsewhere that reported that a lack of physical activity is strongly associated with obesity [34,35,36,37,38,39]. Performing sufficient physical activity might be beneficial to maintain people’s energy expenditure and subsequent energy balance. Therefore, health promotion and education to improve physical activity are required, especially for busy adults in urban areas.

Another important finding of our study is that the prevalence of adult obesity varied across the regencies within the island clusters. The regency with the highest obesity prevalence was Jakarta (28.6%), while the highest among the island clusters was in Java (21.2%). The breakdown of the data reveals that the obese adults in urban areas outnumbered those in the rural areas. This difference can probably be explained by the socio-economic characteristics of each cluster. The Java Island cluster, including Jakarta province, are dominated by people living in urban areas or at least adjacent to urban areas. Meanwhile, urban communities are more likely to have unhealthy lifestyles, such as sedentary behavior and consuming more “unhealthy foods” [40,41]. Meanwhile, in the Sulawesi Island cluster and other island clusters outside the Java cluster, the number and distribution of urban communities is relatively fewer and uneven. A similar finding was reported by a study in Ethiopia that found that men living in metropolitan cities were 1.8 times more likely to become obese than those living in rural areas [36].

The major strength of the present study is that it includes a large sample size. In addition, using weighted factors in the analysis might generate results that more closely represent the Indonesian population. We also acknowledge some limitations in this study. First, the study data were collected cross-sectionally. Therefore, the causality of risk factors and obesity should be cautiously interpreted. Second, we did not disqualify outlier BMI measurement results in the dataset. However, this was only < 1% of the samples, which probably caused a small effect in the analysis. Lastly, the data for high-risk foods were collected using a non-validated questionnaire, which raised response biases of participants’ answers regarding the consumption in the past 30 days. However, since the questionnaire was developed using neutrally worded questions, the options were not led from one to another and did not overlap each other, so that the participants might understand and respond more easily.

## 5. Conclusions

The study implies that there was a variation of obesity prevalence among the provinces and between island clusters. This study provides evidence that obesity risk factors varied across the island clusters, which may have implications in rethinking and redesigning policies and interventions to address the obesity problem in Indonesia. Multiple interventions that address specifically greater risk factors considering cluster characteristics are more likely to be effective in preventing obesity and its negative implications.

## Figures and Tables

**Figure 1 nutrients-14-00971-f001:**
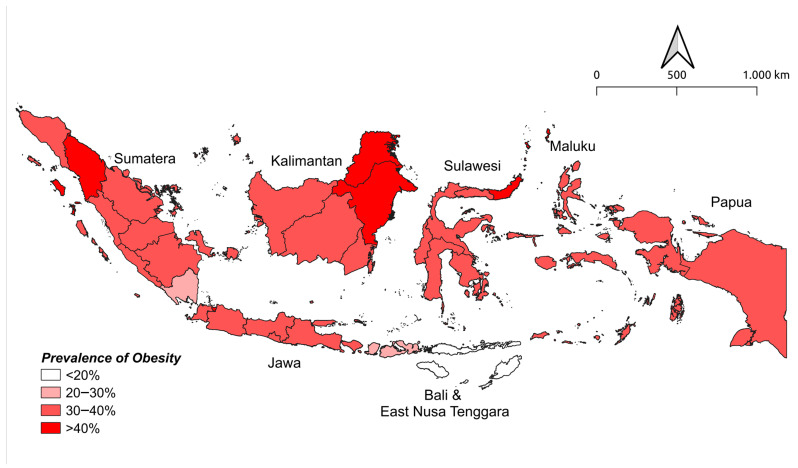
Distribution of obesity prevalence in Indonesia.

**Figure 2 nutrients-14-00971-f002:**
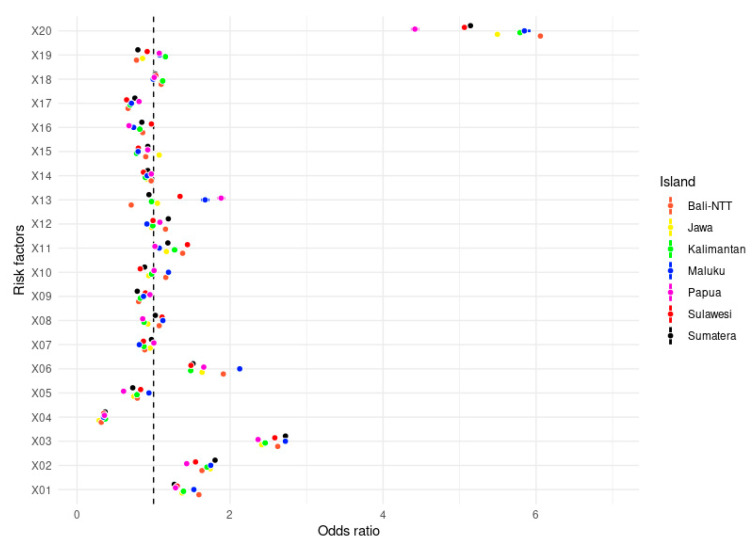
Variation of obesity risk factors by island clusters in Indonesia.

**Table 1 nutrients-14-00971-t001:** Prevalence of obese adults by variables associated with obesity in all island clusters in Indonesia.

Variables	Categories	Island Clusters						
		Sumatra	Java	Bali and Nusa Tenggara	Kalimantan	Sulawesi	Maluku	Papua
		*n* = 180,292	*n* = 204,768	*n* = 50,484	*n* = 59,654	*n* = 85,006	*n* = 18,531	*n* = 20,175
BMI category	Obese	35.3	36.2	28.2	34.8	34.9	35.0	36.0
Location	Rural	31.6	29.9	21.4	29.7	31.6	30.5	32.4
	Urban	40.1	39.6	36.2	40.7	39.9	42.2	43.7
Sex	Men	26.3	26.8	22.3	26.8	26.2	26.8	31.3
	Women	44.6	45.6	33.9	43.6	43.5	43.5	41.2
Marital status	Not Married	18.2	20.6	16.3	19.7	19.7	18.5	22.2
	Married	39.4	39.8	31.4	38.8	39.1	39.2	38.6
	Divorced	33.9	34.2	26.0	30.0	31.0	29.9	40.5
	Widowed	34.5	32.5	22.4	27.8	30.3	35.9	32.8
Age group	≤47 years	34.5	36.8	28.6	35.2	34.7	34.3	35.2
	48–63 years	41.0	40.1	31.9	36.9	39.6	40.4	40.9
	≥64 years	25.2	22.1	16.1	23.3	23.8	26.5	24.5
Education level	Lower education	46.6	46.4	40.1	45.5	43.7	47.2	53.4
	Medium education	36.5	37.9	31.5	37.5	35.8	34.4	39.5
	Higher education	33.2	34.3	24.8	32.3	33.0	33.3	32.2
Occupational status	Unemployed	41.8	42.7	30.1	41.8	40.3	36.5	40.2
	Students	20.2	24.7	15.2	23.1	18.5	19.4	28.7
	Government/civil workers	53.9	54.7	49.3	50.6	52.4	55.4	57.2
	Private company officer	35.0	36.3	34.8	35.9	34.0	38.8	37.3
	Entrepreneur	40.1	42.3	42.9	38.4	41.8	46.9	40.9
	Farmer	25.7	22.9	16.6	23.2	22.4	24.6	28.5
	Fisherman	22.9	24.5	23.9	22.9	25.0	17.4	24.4
	Daily labor/driver/ housekeeper	27.6	27.8	24.1	26.3	29.8	29.8	30.7
	Others	39.9	39.9	28.5	39.0	39.5	42.8	42.8
Mental emotional status	No mental disorder	35.3	36.2	28.8	35.2	35.3	35.8	35.9
	With mental disorder	35.1	36.0	23.6	30.9	31.7	29.5	36.9
Sweet food	<3 times/month	37.0	38.2	26.3	37.6	34.0	34.2	34.3
	1–6 times/week	35.2	35.9	28.8	35.0	34.8	35.3	36.3
	>1 time/day	33.7	35.3	29.2	32.6	35.6	34.7	37.4
Sugar-sweetened beverages	<3 times/month	42.9	44.3	30.4	42.6	39.1	40.5	40.5
	1–6 times/week	36.3	37.5	29.1	36.2	35.4	35.1	34.4
	>1 time/day	30.5	31.2	23.8	30.6	31.8	32.5	37.4
Food high in salt	<3 times/month	36.7	38.0	27.9	36.9	37.7	36.1	36.5
	1–6 times/week	35.0	35.4	28.5	34.1	32.9	34.0	34.6
	>1 time/day	32.4	36.2	28.9	33.9	33.1	35.1	40.5
High-fat food	<3 times/month	34.1	34.4	23.6	32.1	31.0	34.6	32.4
	1–6 times/week	35.4	36.1	29.9	35.3	35.0	35.1	37.8
	>1 time/day	36.5	36.8	32.8	35.7	38.4	35.1	38.2
Meat	<3 times/month	35.3	35.7	27.6	34.9	34.6	34.6	34.8
	1–6 times/week	34.8	37.6	31.4	34.6	36.0	36.5	39.1
	>1 time/day	37.8	36.1	35.2	34.3	34.7	36.1	37.8
Soft or carbonated drinks	<3 times/month	35.9	36.8	28.0	35.7	35.5	35.5	36.1
	1–6 times/week	30.2	31.0	29.6	29.2	31.5	32.3	34.7
	>1 time/day	28.0	32.8	20.2	27.8	35.7	38.6	48.8
Energy drinks	<3 times/month	36.0	36.9	28.8	36.1	36.3	36.3	36.5
	1–6 times/week	26.9	27.2	22.5	25.3	24.9	30.2	33.4
	>1 time/day	26.9	27.2	18.7	24.2	26.5	31.1	37.7
Instant foods	<3 times/month	38.0	37.7	29.6	36.8	37.9	38.8	36.7
	1–6 times/week	33.5	35.2	27.0	34.2	33.5	33.6	35.6
	>1 time/day	31.8	36.3	25.3	26.8	29.2	28.2	35.5
Vegetable and fruit consumption	Sufficient	42.5	42.7	35.3	41.7	37.8	43.2	47.4
	Insufficient	35.0	35.9	27.8	34.5	34.7	34.4	35.2
Smoking	Never smoked	41.7	43.4	32.7	40.9	41.5	41.2	39.4
	Quit smoking	37.1	37.2	30.9	37.6	34.0	47.2	39.1
	Currently smoking	24.0	23.5	19.1	22.7	22.9	23.7	28.7
Physical activity	Adequate	32.2	32.4	24.6	29.9	31.1	31.9	33.0
	Not adequate	37.8	39.0	31.0	38.4	37.6	36.8	38.9
Alcohol consumption	Yes	28.4	27.2	24.5	20.7	25.1	23.6	29.2
	No	35.5	36.4	28.7	35.6	35.9	36.6	36.5
Blood pressure	Normal	22.3	21.5	16.4	19.9	21.6	22.5	24.1
	Pre-hypertension	34.6	34.2	28.7	31.4	35.2	35.4	36.3
	Hypertension stage 1	46.1	45.1	38.5	43.8	44.9	45.9	49.2
	Hypertension stage 2	54.7	52.4	44.8	51.2	51.3	55.9	58.5

## Data Availability

The datasets generated and analyzed for this study can be found via https://www.litbang.kemkes.go.id/ (accessed on 3 May 2021) through a request process at the Institute of Health Research and Development of the Indonesian Ministry of Health.

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
