# Peer review of "Obesity Risk-Factor Variation Based on Island Clusters: A Secondary Analysis of Indonesian Basic Health Research 2018"

_nutrients, 2022, doi:10.3390/nu14050971_

Round 1
Reviewer 1 Report
This work requires major changes. Comments and suggestions are attached.
The study addresses an important issue related to the determinants of obesity and the need to clearly identify them for proper implementation of public policies and effective government programs. On the other hand, the study involves a large number of subjects.
However, this study has weaknesses that could bias the results regarding the impact of different determinants on obesity in this population. It covers a wide age range that includes subjects who are not adults because they are in adolescence (population from 15 to 18 years old) to subjects in emerging adulthood, adults and older adults. Lifestyles are usually very different in young subjects and adolescents when compared to older subjects. Lifestyles are usually very different in young subjects and adolescents when compared to older subjects. This study included subjects from 15 years of age who were analyzed together with the group under 47 years of age. When performing this analysis in this group with a wide age range, there may be a bias in the results due to the great variability of some sociodemographic factors associated with age.
It is not clear why obesity was considered from a BMI of 27.5 as many for subjects younger than 17 years as for those older than that age.
Although the statistical analysis allows us to estimate the influence of the different determinants , it does not allow us to see the size of the differences as a way of seeing which factors really have a greater influence on the prevalence of obesity, which would allow us to estimate which factors should be privileged in the government policies and programs.
I believe that major changes are necessary, which allow for an analysis by age groups, separating adolescents and subjects in emerging adulthood from adults and older adults in order to adequately identify the main determinants of obesity in each place. In the statistical analysis, methodologies that allow estimating the size of the differences (Cliff's delta and Cohen's delta) should be considered.
The discussion fails to adequately substantiate the results found, either in relation to the variability between the different islands or those that were controversial when compared with other studies.
Author Response
Response to Reviewer 1 comments on nutrients-1580409
Titled: “Obesity Risk Factor Variation Based on Island Clusters: A Secondary Analysis of Indonesian Basic Health Research 2018”
Dear Reviewer 1,
On behalf of the authors, I would like to thank you for your email informing the results of reviewer evaluation for our manuscript entitled "Obesity Risk Factor Variation Based on Islands Cluster: A Secondary Analysis of Indonesian Basic Health Research 2018" (nutrients-1580409).
I also want to thank the reviewers for their valuable and helpful comments to improve the paper's quality. Based on the reviewer concerns, we have made clarifications and changes in methods, results, and discussion sections. In detail, we put the point-by-point responses to the comments as changes appeared in the manuscript main-text. In this letter, I would like to take this opportunity to express our sincere thanks to the reviewers who identified areas of our manuscript that needed clarifications or modification. We would also like to thank you for allowing us to resubmit this manuscript.
Should you have further questions or concerns about our manuscript, please do not hesitate to let us know. We look forward to hearing the review outcome of this manuscript.
Sincerely,
Sri Astuti Thamrin
Responses to reviewer’ comments
Reviewer 1.
This work requires major changes. Comments and suggestions are attached.
The study addresses an important issue related to the determinants of obesity and the need to clearly identify them for proper implementation of public policies and effective government programs. On the other hand, the study involves a large number of subjects.
Response: Dear reviewer, thank you very for much for giving insight and valuable feedback in order to improve the quality of this paper. We have made some changes regarding your comments.
However, this study has weaknesses that could bias the results regarding the impact of different determinants on obesity in this population. It covers a wide age range that includes subjects who are not adults because they are in adolescence (population from 15 to 18 years old) to subjects in emerging adulthood, adults and older adults. Lifestyles are usually very different in young subjects and adolescents when compared to older subjects. Lifestyles are usually very different in young subjects and adolescents when compared to older subjects. This study included subjects from 15 years of age who were analysed together with the group under 47 years of age. When performing this analysis in this group with a wide age range, there may be a bias in the results due to the great variability of some sociodemographic factors associated with age.
Response: We do re-analysis with the data exclude those aged 15-18 years. Besides physiological and psychological differences between adolescents and adults that may affect factors associated with obesity, the indices to determine these age groups are also different (BMI vs BMI-for-age-z-score). Therefore, we decided to include exclude adolescents (line 78-80).
It is not clear why obesity was considered from a BMI of 27.5 as many for subjects younger than 17 years as for those older than that age.
Response: We have made a revision for the cut-off of obesity using >=25 kg/m2 following the WHO Asia-pacific standard. Please see lines 26, 89-91.
Although the statistical analysis allows us to estimate the influence of the different determinants, it does not allow us to see the size of the differences as a way of seeing which factors really have a greater influence on the prevalence of obesity, which would allow us to estimate which factors should be privileged in the government policies and programs.
I believe that major changes are necessary, which allow for an analysis by age groups, separating adolescents and subjects in emerging adulthood from adults and older adults in order to adequately identify the main determinants of obesity in each place. In the statistical analysis, methodologies that allow estimating the size of the differences (Cliff's delta and Cohen's delta) should be considered.
Response: We have added Cliff’s delta and Cohen’s delta in lines 131-135, lines 188-189, line 316.
The discussion fails to adequately substantiate the results found, either in relation to the variability between the different islands or those that were controversial when compared with other studies.
Response: We have described variability of risk factors across the clusters in paragraph 5-7 (line 225-255).
Reviewer 2 Report
The manuscript on the variation of the risk factor for obesity according to the group of islands in Indonesia is an interesting topic, due to the increase that has occurred in obesity in recent decades and the need to establish prevention and intervention strategies. However, the document needs major changes:
- In the introduction, the main objective of the study was to examine the factors that contribute to obesity in adults and to investigate how they vary between age groups. The definition of obesity is not adequate. It calls children between 15 and 18 years old an adult and defines their obesity with a cut off, instead of using percentiles. The cut off of 27.5 does not respond to any international criteria. They should repeat the study with an adequate definition of obesity.
- In the material and methods section, it is indicated that there is detailed information on methods, ethical considerations and other aspects related to the RISKESDAS study. However, the bibliographic citation is not well referenced, so these data cannot be confirmed. They must cite correctly in order to review said information.
- In the same section, the frequency of dietary risk foods, sweet foods and drinks, high-salt foods, high-fat foods, meats, carbonated drinks, energy drinks, and instant foods were measured. Eating or drinking risk foods more than once a day was considered high frequency. How were these answers validated? What questionnaire was obtained for it? Was the consistency of the answers evaluated?
- Throughout the text, the authors speak of variables that are the cause of obesity, it would probably be appropriate to speak of the association between certain variables and the greater or lesser prevalence of obesity.
Author Response
Response to Reviewer 2 comments on nutrients-1580409
Titled: “Obesity Risk Factor Variation Based on Island Clusters: A Secondary Analysis of Indonesian Basic Health Research 2018”
Dear Reviewer 2,
On behalf of the authors, I would like to thank you for your email informing the results of reviewer evaluation for our manuscript entitled "Obesity Risk Factor Variation Based on Islands Cluster: A Secondary Analysis of Indonesian Basic Health Research 2018" (nutrients-1580409).
I also want to thank the reviewers for their valuable and helpful comments to improve the paper's quality. Based on the reviewer concerns, we have made clarifications and changes in methods, results, and discussion sections. In detail, we put the point-by-point responses to the comments as changes appeared in the manuscript main-text. In this letter, I would like to take this opportunity to express our sincere thanks to the reviewers who identified areas of our manuscript that needed clarifications or modification. We would also like to thank you for allowing us to resubmit this manuscript.
Should you have further questions or concerns about our manuscript, please do not hesitate to let us know. We look forward to hearing the review outcome of this manuscript.
Sincerely,
Sri Astuti Thamrin
Responses to reviewer’ comments
Reviewer 2.
The manuscript on the variation of the risk factor for obesity according to the group of islands in Indonesia is an interesting topic, due to the increase that has occurred in obesity in recent decades and the need to establish prevention and intervention strategies. However, the document needs major changes.
Response: We thank the reviewer for the valuable suggestion. We have modified and changed based on your suggestions.
In the introduction, the main objective of the study was to examine the factors that contribute to obesity in adults and to investigate how they vary between age groups. The definition of obesity is not adequate. It calls children between 15 and 18 years old an adult and defines their obesity with a cut off, instead of using percentiles.
Response: We have revised based on your suggestions excluding those 15-18 years (line 78-81).
The cut off of 27.5 does not respond to any international criteria. They should repeat the study with an adequate definition of obesity.
Response: We have revised based on your suggestions using the WHO BMI cut-off standard for Asian population. Please see lines 89-91.
In the material and methods section, it is indicated that there is detailed information on methods, ethical considerations and other aspects related to the RISKESDAS study. However, the bibliographic citation is not well referenced, so these data cannot be confirmed. They must cite correctly in order to review said information.
Response: We made revisions based on your suggestion. We cite RISKESDAS report in the reference section. Subheading of 2.5. Ethical consideration was added in the manuscript. Please see line 129-131.
In the same section, the frequency of dietary risk foods, sweet foods and drinks, high-salt foods, high-fat foods, meats, carbonated drinks, energy drinks, and instant foods were measured. Eating or drinking risk foods more than once a day was considered high frequency. How were these answers validated? What questionnaire was obtained for it? Was the consistency of the answers evaluated?
Response: Thanks for giving critical feedback regarding high-risk food intakes. We fully understand that there are potential biases from participants’ recalls, especially because the questions required closed-ended answers about what they consumed 30 days back. In this case, we were unable to make modifications to the data collection method procedure for the existing data, particularly for dietary data. Hence, we put this issue as a study limitation. Please see line 266-270.
Throughout the text, the authors speak of variables that are the cause of obesity, it would probably be appropriate to speak of the association between certain variables and the greater or lesser prevalence of obesity.
Response: We made additional changes in the discussion section to highlight the variation among island clusters.
Round 2
Reviewer 1 Report
The author has appropriately accommodated the recommendations made by the reviewers. They have also recognized some weaknesses in the discussion that could have biased the results to some extent
Author Response
Response to the Reviewer 1 21 Feb. 22
The Authors need to clearly state in their manuscript if the subjects provided a written consent form prior to their participation in the study.
Response:
Dear Reviewer 1, we have added the information in the informed consent section. Please see line 324-325.
